# Hydrogels from the Assembly of SAA/Elastin-Inspired Peptides Reveal Non-Canonical Nanotopologies

**DOI:** 10.3390/molecules27227901

**Published:** 2022-11-15

**Authors:** Alessandra Scelsi, Brigida Bochicchio, Andrew M. Smith, Antonio Laezza, Alberto Saiani, Antonietta Pepe

**Affiliations:** 1Laboratory of Bioinspired Materials (LABIM), Department of Science, University of Basilicata, 85100 Potenza, Italy; 2Department of Materials, Manchester Institute of Biotechnology, Faculty of Science and Engineering, The University of Manchester, Manchester M13 9PL, UK

**Keywords:** self-assembling peptides, oscillatory rheology, atomic force microscopy, nanotopology

## Abstract

Peptide-based hydrogels are of great interest in the biomedical field according to their biocompatibility, simple structure and tunable properties via sequence modification. In recent years, multicomponent assembly of peptides have expanded the possibilities to produce more versatile hydrogels, by blending gelating peptides with different type of peptides to add new features. In the present study, the assembly of gelating P5 peptide SFFSF blended with P21 peptide, SFFSFGVPGVGVPGVGSFFSF, an elastin-inspired peptides or, alternatively, with FF dipeptide, was investigated by oscillatory rheology and different microscopy techniques in order to shed light on the nanotopologies formed by the self-assembled peptide mixtures. Our data show that, depending on the added peptides, cooperative or disruptive assembly can be observed giving rise to distinct nanotopologies to which correspond different mechanical properties that could be exploited to fabricate materials with desired properties.

## 1. Introduction

The use of noncovalent molecular self-assembly to produce nanostructured materials has become a new strategy for the construction of functional nanobiomaterials useful in various areas such as physics, chemistry, biology, nanoscience, and materials science [1,2,3].

Taking inspiration from nature that uses self-assembly for the building of numerous biological structures, many researchers focused their attention on self-assembling peptides as versatile building blocks for fabricating materials with defined nanoarchitecture by a bottom-up approach. Various molecular designs have been developed for the synthesis of self-assembling peptides, with the main families being amphiphilic peptides, α-helix, coiled−coil peptides, and β-sheet peptides [4,5,6,7,8]. A wide range of this research is dedicated to small-sized peptides that have the ability of self-assembling into supramolecular β-sheet fibrils and this is an increasingly utilized route for the synthesis of new nanostructured materials [9,10]. Moreover, peptide self-assembly in β-sheet fibrils is extensively utilized to produce hydrogels, hydrated materials composed of dilute networks of nanofibers [11,12]. Recent studies were able to define some insights in the parameters that allow control over fiber morphology and stiffness, hydrogel topology, and mechanical properties [13]. For example, the interactions between individual fibers increases by varying the amino acid sequence, such as the inclusion of more hydrophobic amino acids, or by varying pH media, or through the addition of salt when charged amino acids are present [13]. On the other side, because the self-assembled structures are composed of many copies of simpler molecular components, they can be highly modular and easily modified. For these reasons the material properties can be readily tuned by changing the mixture of the molecules that make them up, and great control of length scales, aspect ratios, and higher-order hierarchical self-assembly may be achieved thanks to modularity of the supramolecular structures [9].

To expand the options in producing materials with tailored properties, alternative and more flexible strategy were developed with investigation of co-assembly of different peptides [14,15,16,17,18]. The addition of bioactive sequences to the self-assembling peptide was widely exploited in order to improve interaction of the hydrogels with biological systems. Co-assembly of the self-assembling peptide with the functionalized peptide in various ratios results in a hydrogel scaffold that elicited bioactivity [19,20].

Recently, we demonstrated that amyloid-like SA5N peptide of sequence NH_2_-SFFSF-NH_2_ self-assembles into biocompatible hydrogel suitable for cell growth. Additionally, we found that mechanical properties of these hydrogels can be tuned using longer peptide SA21 that changed elasticity of scaffolds for improved cell settlement [21].

In the present study, we investigated self-assembled nanostructures obtained by the multicomponent assembly in different ratios of three SAA-related aromatic peptides (P5, P21, FF), varying both in the ratio of aromatic moieties and in the length of peptides.

The explored peptides are self-assembling peptides of sequence NH_2_-SFFSF-OH (P5), the above mentioned NH_2_-SFFSFGVPGVGVPGVGSFFSF-OH (P21), and NH_2_-FF-OH (FF) (Figure 1). P5 peptide corresponds to the hSAA_2-6_ sequence in the N-terminal region of human serum amyloid A protein (hSAA1) previously shown to form amyloid-like nanofibers [22]. Furthermore, recent studies on truncated peptides of hSAA1 have shown that the spontaneous amyloid fiber formation of P5 grew together with microcrystals [22]. Furthermore, Westermark has shown the prompt formation of SAA truncated peptides into amyloid fibers in 10% acetic acid (AcOH) [23]. The P21 peptide is constituted by two external sequences of P5 as self-assembling units connected by an elastin-inspired linker sequence in the central part. The design of the three-block peptide contained a central elastin-inspired sequence that ensures high flexibility to the hinge region and enables the peptide to arrange in nanospheres [24].

In this report, we investigated the production of a series of hydrogels obtained by the multicomponent assembly of P5 peptide with P21 or FF peptides at variable ratios. The morphology of the nanostructured hydrogels was investigated by atomic force microscopy (AFM), scanning electron microscopy (SEM), and transmission electron microscopy (TEM), showing different nanotopologies arising from the assembly. Furthermore, the mechanical properties were measured by dynamic oscillatory rheology and correlated to the observed nanostructures.

Amyloid fibers are commonly characterized by linear unbranched structures that could assume tape, ribbon, or helical structures. Accordingly, these are considered as canonical nanostructures of amyloid peptides. Our data show that, according to the added peptides, cooperative or disruptive assembly can be observed giving rise to distinct non-canonical nanotopologies, such as tree-like structures, necklaces, or vesicles. The multicomponent hydrogels show different mechanical properties that could be exploited to fabricate materials of desired properties.

## 2. Results and Discussion

### 2.1. P5 Peptide Assembly

The P5 peptide was investigated by CD spectroscopy in order to define the secondary structure (Appendix A). Unfortunately, the CD spectra were of limited utility, being highly dominated in the far UV region by aromatic contributions (Appendix A). This has been observed for other diphenylalanine peptides, such as KLVFF and correlated peptides [25,26]. The importance of aromatic contribution emerged also from the near UV spectrum (Appendix A), where three bands in the 250–270 nm region are recorded. These bands arose from electronic transition of the aromatic ring and witnessed a conformational restriction of the rotation of the side chain, due to self-assembly and probable π−π aromatic interactions.

Peptides derived from the N-terminus of hSAA were previously characterized for their amyloid formation propensity by SEM microscopy, Fourier transform infra-red (FT-IR) spectroscopy and thioflavin T (ThT) assay [22]. The results showed that all the investigated peptides, comprising hSAA_2-6_ corresponding to P5 peptide, show typical amyloid signatures, such as β-sheet structures and a left-handed helicity of the twisted fiber structures [22]. It was suggested that the presence of three phenylalanines in its short sequence, was able to stabilize the β-sheet arrangement by forming the so-called Phe ladder. [27] Our CD data confirm this hypothesis.

Even if a previous report described the formation of amyloid fibers from hSAA_1-12_ peptide and related truncated peptides, studies on the formation of hydrogels on these peptides were lacking [23]. Accordingly, we defined the experimental conditions and the mechanical properties of hydrogels formed by the short P5 peptide. At variance with previous studies where the peptides were studied in 10% (*v*/*v*) acetic acid (AcOH), the gelation of P5 peptide was performed in 5% (*v*/*v*) DMSO. First, the macroscopic behavior of the P5 samples prepared at different concentrations was investigated by the vial inversion test. The hydrogel formation was detected visually by inverting the vial and considered successful if the hydrogels did not collapse. The same approach was employed for hydrogels obtained by the multicomponent assembly of P5 and P21 peptides. The studied hydrogels are described in Table 1.

### 2.2. Oscillatory Rheology

The mechanical properties of the H1, H2, H3 hydrogels composed of different concentrations of P5 peptide were studied using oscillatory rheology. Strain amplitude sweeps were performed for each sample to determine the linear viscoelastic region (LVR) (Appendix A).

Then, the storage and loss moduli as a function of radial frequency were recorded as shown in Figure 2 for different concentrations of P5. Each examined concentration shows a storage modulus G’ that is weakly dependent on the frequency, a feature typical of cross-linked polymeric hydrogels. However, since the P5 networks are not chemically cross-linked, this observation indicates significant attractive interfibril interactions that physically cross-link the fibrils [28]. Furthermore, the value of G’ is always higher than G’’ for these hydrogels, indicating that the samples behave as a solid but with different elasticity. When the storage, G’, and loss, G’’, moduli obtained for the P5 sample series were represented as a function of concentration, an increase in both moduli is observed due to a higher peptide concentration, forming stronger hydrogels. The predicted scaling behavior for these samples can be tentatively extrapolated by fitting the G’ data at different concentrations to the power law model, G’∞C^α^, with α value of 5.1 (Figure 2b). The elasticity of these hydrogel systems can be often successfully described using theories developed for flexible and semiflexible polymers [29,30], with fiber rigidity and network mesh size being the prevalent parameters.

According to these models, the scaling α value for densely cross-linked gels is 2.5, (G’∞ c ~α^5/2^), but a higher value of 3.1 is reported by Miller et al. for the FEKII18 system, ascribed to the increase in relatively short network junction points, reducing persistence length of the fibers and decreasing mesh size observed for the self-assembled peptide system [13].

However, fiber–fiber interactions of different natures could also affect the stiffness of the network. A scaling α value of 5.1 is particularly high and could be assigned to a network with a high number of junction points and smaller and smaller mesh sizes with increasing of peptide concentration or to highly packed fibers that could laterally aggregate and become more rigid in a hierarchical assembly. This would lead to a higher concentration dependence of G’. Similar observations are described by Rüter et al. studying other amyloid-like systems constituted by A8K and A10K peptides [31].

The concentration dependence of the elastic modulus in low-frequency regime provides a helpful way to highlight physical aspects of hydrogels on the microscopic scales that could be ascribed to the architecture of fiber molecules, their arrangement, and their interactions [32].

### 2.3. Supramolecular Nanostructures of Hydrogel as a Function of Concentration

In order to get deeper insight into the supramolecular structure of the hydrogels, the samples were observed by different microscopy techniques. AFM and SEM have provided strong supports for investigating the nanofibers formed by self-assembling P5 peptide. AFM gives 3D images with 10 nm resolution and without perturbing the samples. The same samples can be used to get the SEM analysis and confirm the sample’s morphology at lower scale, evidencing increasing hierarchical levels of aggregation.

In Figure 3, images of AFM and SEM microscopies of H1, H2, and H3 hydrogels are shown, with clear evidence of more densely packed fibers in presence of increasing P5 concentration. AFM images of H1 (Figure 3a, top) show that the aligned fibers are organized in bundles. Each bundle is composed by a high number of fibers. In Figure 3b, the AFM image of H2 shows P5 fibers that associate in a higher number, while in Figure 3c, the AFM image of H3 shows high densely packed bundled fibers, with smaller fibers coming out underneath. The surface’s characterization by SEM confirms, in Figure 3a at the bottom, the presence of the bundles of fibers. In these fibers the presence of periodicity is not appreciable, but they form and maintain a three dimensional structure, as confirmed by SEM images in Figure 3b,c at the bottom. Moreover, the surface morphology of these hydrogels are more packed with increasing of the concentration, as issued in the SEM images, revealing a dense film at the highest concentration with an emerging fiber core. Single-fiber TEM images were recorded on diluted samples, with the aim to point out more defined morphology of the single fibers. H1 hydrogel (Figure 4a) shows tiny fibers with a diameter of 14.8 ± 2.7 nm, with different lengths and directionalities. Some are straight while many are bending, creating an intricate meshwork. H2 hydrogel (Figure 4b) has fibers with larger diameters (59.2 ± 20.5 nm) coming out from the alignment and interactions of more fibers. Additionally, the direction is different, being almost all linear. Fiber morphology observed for H3 hydrogel (Figure 4c) is constituted by straight larger fibers and some very tiny bending fibers that interlock them in a subtle network as discernible at higher magnification (Appendix A).

### 2.4. Assembly of P5 and P21 Peptides

To investigate the impact of P21 peptide on the mechanical properties of the hydrogel network topologies, the elastic, G’ and viscous, G’’ moduli were measured for hydrogels obtained from mixtures of P5 and P21 using oscillatory rheology (Figure 5).

The G’ value of all the hydrogels (H4–H11) with different concentrations of P21 is always higher than G’’ (Figure 5a,b). It is noticeable that the value of the elastic modulus G’ increases slightly in the analyzed angular frequency range. Comparing the rheological data as a function of P21 concentration, the following observations are drawn (Figure 5b). For P5 1% (*w*/*v*), the G’ modulus is 914.4 Pa and it remains approximately constant in presence of P21 0.017% (*w*/*v*). After that, the presence of P21 0.034% (*w*/*v*) induces a jump of the modulus G’, to 8 kPa. Further addition of P21 (0.084%, *w*/*v*) increments the G’ value to 13.5 kPa, while doubling the concentration of P21 to 0.17% *w/v* does not significantly change the elastic modulus of the hydrogel reaching a plateau region. Similar comments can be reported for the influence of P21 in the P5 2% (*w*/*v*) hydrogels (Appendix A). In fact, in the absence of P21, the G’ of P5 2% hydrogel is 13 kPa and is not significantly different from the G’ value of P5 2% (*w*/*v*) with 0.034% (*w*/*v*) P21. A further increase in P21 concentration, 0.068% (*w*/*v*), induces a jump of G’ (39 kPa), while for P21 0.17% (*w*/*v*) G’ is 39.5 kPa, and it remains approximately constant at 0.34% (*w*/*v*) of P21, taking to a plateau region.

The data indicate that P21 plays a significant role in tuning the mechanical properties of P5 hydrogels. Hence, an increase in the P21 amount at specific P21/P5 molar ratio determines a jump of G’ with an significant change of its value. It might be that the P21 elastin-inspired peptide sequence introduces the possibility of movements among the fibers of P5 and induces strong dependence of elastic modulus from the P21 concentration. After this increase, the addition of P21 results in a plateau, suggesting that the system obtained stability because the formed entanglements are impediments that interfere with the reptation movements and with the P5 fibers relaxation.

The comparison of the two mixture of P5 1% and 2% with P21 showed marked differences in G’ value and point out that the increase in the strength of the hydrogels was due to the change in the hydrogel structure in the presence of P21.

However, a notable aspect that emerged from the data is that although different concentrations of P5 have been used, 1% and 2% (*w*/*v*), the trend is comparable for constant molar ratios of P21 (Appendix A). The sharp increase in G’ observed for the same P21/P5 molar ratio suggests that there could be a precise balance among opposite features.

By AFM imaging of the hydrogel differences in the surface morphology were observed (Figure 6). However, the high density of the samples did not reveal the intimate morphological features of the constituent nanostructures.

In order to highlight differences in the supramolecular structures of the multicomponent peptide hydrogel, also TEM images of the hydrogels were acquired on diluted samples. Even if we were conscious that by dilution the representation of the actual network could be lost, we investigated with some caution the diluted samples in order to get the lower level of the hierarchical assembly with the aim to show the subtle/fine morphology of the constituting nanofibers. Nanofibrous structures were observed in all the samples (Figure 7); however, the differences emerged when compared to the morphologies observed by TEM for the monocomponent P5 hydrogels (Figure 4), thus underlining the determining effects that the P21 peptide has on the nanostructure of the resulting hydrogels. Long nanofibers are observed for H4 (Figure 7a), which entangle in few points; more entanglements, together with some forked and bundling fibers are discernible in H5 (Figure 7b), while tree-like organization with short branches is observed for H6 (Figure 7c). P21 peptide seems to introduce branching points across the fibers of P5 hydrogels, yielding a structure with forked fibers that further evolve into branching fibers at higher P21 amounts (Figure 7c), as the TEM images suggest. Probably, the SFFSF sequence present at the terminals of the P21 sequence could be able to interact with the fiber-forming P5, while the elastic GVGVPGVGVPG sequence departing from the fibers could promote the growth of a new branch of fibrils. Considering that the concentration of P5 is not varied, a higher number of branching points probably induces a lowering of the length of the fibrils. The ratio of P5:P21 of 100:1 could represents the critical value, where a number of branches giving rise to a higher number of junction points are balanced by the sufficient length of the fibrils concurring with the entanglements (Figure 8). Further studies are required to confirm this speculation.

### 2.5. Alternative Nanotopologies of Peptide Assembly in 10% Acetic Acid (AcOH)

Previous studies have shown that P5 in 10% AcOH self-assembled into left-handed twisted rope amyloid fibers as well as into microcrystals [22]. We further observed that P5 peptide (1% *w/v* in 10% AcOH, *v/v*) gives rise to a hydrogel (inversion test) with lower storage modulus than the hydrogel formed in 5% DMSO (*v/v)* (Appendix A). Additionally, the assembly of P5 and P21 in 10% AcOH (*v/v )* formed weak hydrogels (data not shown). In order to investigate the morphology of the self-assembled structures, AFM images on the assembled P21/P5 hydrogels were acquired (Figure 9). To observe the fine structure, the hydrogels were diluted ten-fold. The AFM images of P5 peptide shown in Figure 9a highlight the presence of intertwined fiber structure. At higher magnification, the presence of twisted rope fibers together with untwisted fibers is observed (Appendix A).

The assembly with P21 in 10% AcOH (*v*/*v*) changes significantly the observed nanotopology (Figure 9b,c). Fibers are still present but the presence of vesicular structures with fibers departing from them are evident. The dimension of the vesicles is related to the concentration of P21, with higher concentration of P21 showing globular structures of larger diameter (Figure 9c). P21 was previously shown to form nanospheres with a mean diameter of 150 nm in HFIP 50% (*v*/*v*) solution [24]. The formation of nanospheres was suggested to be promoted by π−π stacking interactions of the Phe residues that induced a hairpin structure assembly of the P21 peptide. The formation of flexible loop structures is based on the arrangement of the SAA sequences of P21 inside the core and the elastin sequences as the corona of the nanospheres. We suppose that also in the investigated solution conditions (10% AcOH, *v*/*v*), the P21 peptide is organizing in spherical structures while the P5 peptides are forming fibers departing from the corona of the spherical structures. Similar structures in a necklace fashion were observed in the multicomponent assembly of other amyloid peptides, suggesting an alternative way of integration of the spherical and fibrous structures [33].

### 2.6. Assembly of SA5 and FF Peptides in Different Ratios

FF dipeptide is well known for its self-assembly in nanotubes of high stiffness [34,35]. Even if FF dipeptide is not able to form hydrogel, we assumed that it could change the mechanical properties of the P5 hydrogel. The presence of FF dipeptide sequence in P5 sequence infers possible interactions between the peptides that could modulate the hydrogel formation. We investigated the P5 hydrogels supplemented with FF monomers and nanotubes (Table 2). The first attempts with FF dipeptide added to the 1% P5 gelation solution at a final concentration of 0.025% (*m/v*) hindered the gelation. Probably the FF peptides are able to recognize P5 peptide, thus hindering the fiber formation. Similar results were observed for Aβ amyloid assembly, where a KLVFF pentapeptide is able to inhibit the amyloid fiber formation [36,37]. In a second attempt, FF nanotubes were pre-formed in aqueous solution and subsequently added to the P5 gelation solution. The obtained hydrogels were investigated by oscillatory rheology and scanning electron microscopy.

In Figure 10a, the G’ and G’’ moduli recorded for P5 hydrogels containing increasing concentrations of FF nanotubes are shown. As evident from the trend of G’ as a function of FF concentration (Figure 10b), the addition of preformed FF nanotubes weakened dramatically the hydrogels. Scanning electron microscopy images suggested self-sorting as the main mode of association inferring a disruptive assembly (Appendix A). Both the rheological data and the SEM microscopy images ruled out the occurrence of interactions among FF nanotubes and P5 during hydrogel self-assembly. This data suggested that even if the P5 sequence has a FF core, this is not sufficient for a cooperative assembly with the FF dipeptide. Probably, the presence of the two hydrophilic serine residues in the P5 peptide hindered interactions by shielding the hydrophobic Phe aromatic side-chain of the FF nanotubes.

## 3. Materials and Methods

### 3.1. Peptide Synthesis

The peptides were synthesized by solid phase peptide synthesis (SPPS) with a Tribute automatic peptide synthesizer (Protein Technologies Inc., Tucson, AZ, USA), using Fmoc/tBu chemistry. Fmoc-α-aminoacids were purchased from Inbios (Pozzuoli, Italy), and coupling reagent HBTU was acquired from Matrix Innovation (Quebec, QC, Canada). Reagent used for cleavage of the peptides from resin was an aqueous solution of 95% trifluoroacetic acid, (TFA, >99%). The peptides were purified by semipreparative reversed-phase HPLC, using binary gradient H_2_O (0.1% TFA) and CH_3_CN (0.1%TFA) as solvents. The purity of the peptides was assessed by HRMS (ESI) mass spectrometry (P21 *m*/*z* 1063.52966 [M+2H]^2+^, 1163.52679 calcd. for [C_106_H_144_N_21_O_26_]^2+^), (P5 *m*/*z* 634.28717 [M+H]^+^, 634.28714 calcd. for [C_33_H_40_N_5_O_8_]^+^), (FF *m*/*z* 313.15497 [M+H]^+^ 313.15467 calcd. for [C_18_H_21_O_3_N_2_]^+^, and ^1^H NMR spectroscopy (Appendix A).

### 3.2. CD Spectroscopy

Circular dichroism spectra were recorded on a JASCO J-815 Spectropolarimeter (JASCO, Milan, Italy), equipped with HAAKE temperature controller. For far-UV analysis P5 sample was dissolved at a concentration of 0.01% *w*/*v* in H_2_O/HFIP (1/1, *v*/*v*)) and loaded into cylindrical quartz cells with pathlengths of 1 mm. Spectrum was acquired in a wavelength range of 190–250 nm at room temperature, with a scan speed of 20 nm/min and a band width of 1 nm. CD spectra represented the average of 9 scans and all final spectra were obtained after subtracting the background. Data are expressed in terms of [Θ] and the molar ellipticity in units of degree cm^2^ dmol^−1^. For near–UV spectrum P5 sample was dissolved at a concentration of 0.1% *w*/*v* in H_2_O/HFIP (1/1, *v*/*v*)), loaded into cylindrical quartz cells with pathlengths of 1 mm. The CD spectrum was recorded in the range 250–360 nm in the 1 mm pathlength quartz cell.

### 3.3. Hydrogel Preparation

Hydrogels at a final concentration of 0.71%, 1.0%, and 2.0% (*w*/*v*) were prepared by dissolving the weighed P5 peptide powder in 10 μL of DMSO and 200 μL of MilliQ water or in 10% acetic acid (AcOH). The solution was mildly agitated (5 min) at room temperature and then left to gel for 24 h.

Hydrogels composed of a blend of P5 and P21 were prepared by adding P21, previously dissolved in DMSO to P5, followed by MilliQ water. The mixture was mildly agitated (5 min) at room temperature and then left to gel for 24 h.

Hydrogels composed of a blend of P5 and FF were prepared by adding FF, previously dissolved in DMSO to P5, followed by MilliQ water. The mixture was mildly agitated (5 min) at room temperature and then left to gel for 24 h.

Hydrogels composed of a blend of P5 (1% *w*/*v* final concentration) and preformed nanotubes of the dipeptide FF at 65° in MilliQ water were prepared dissolving the P5 powder in 10 μL of DMSO, followed by the MilliQ water. The mixture was mildly agitated (5 min) at room temperature and then left to gel for 24 h.

FF nanotubes were prepared by a procedure described in [38]. Briefly, 10 mg of lyophilized peptide (L-Phe-L-Phe) was dissolved in 5 mL of MilliQ water at 65 °C, and the sample was equilibrated for 30 min and then gradually cooled to room temperature

### 3.4. Oscillatory Rheology

Rheological studies were performed on TA instrument AR-G2 rheometer equipped with a Peltier device to control the temperature. Parallel plate geometry with 20 mm diameter with a 0.250 mm gap was used. The samples were prepared as described in Section 3.2 and after 24 h of gelation was poured on the plate. To minimize the evaporation a solvent trap was applied. Initially, strain amplitude sweeps (γ = 0.04–10%) were performed at constant angular frequency (ω = 1 rad s^−1^) to identify the linear viscoelastic region (LVR). Subsequently, frequency sweeps (ω = 0.06–94.2 rad s^−1^) at constant strain amplitude (γ = 0.2%) were performed to determine the elastic (G′) and viscous (G″) moduli within the LVR. All measurements were performed at 25 °C, and all measurements were repeated at least three times to ensure reproducibility.

### 3.5. Atomic Force Microscopy (AFM)

A 10 μL aliquot of the sample was deposited onto a silicon wafer on Silicon (100) wafer substrate (Aldrich, Saint Louis, MO, USA). The silicon wafers were cleaned by using a two-solvent method consisting in the immersion of the Si wafer in warm acetone bath for a period of 10 min. Then a methanol bath for a period of 5 min immediately followed with final deionized water rinses. Finally, they were air-dried prior to imaging. Samples were stored in a Petri dish until observed by the scanning force microscope (Park Autoprobe XE-120, Park Systems, Suwon, Korea). In some cases, samples were prepared by diluting the hydrogels tenfold to highlight the fibrous structures. Specimens were observed at room temperature. Data acquisition was carried out in intermittent contact mode at scan rates between 0.4 and 3 Hz, using rectangular Si cantilevers (NCHR, Park Systems, Suwon, Korea) having the radius of curvature less than 10 nm and with the nominal resonance frequency and force constant of 330 kHz and 42 N m^−1^, respectively.

### 3.6. Transmission Electron Microscopy (TEM)

Hydrogels were diluted by a factor of 10 with MilliQ water. A carbon-coated copper grid (400 mesh) was then placed sequentially on 10 μL sample for 1 min, 10 μL ddH_2_O for 10 s, 10 μL 1% uranyl acetate solution for 30 s, and 10 μL of ddH2O for 10 s. Excess liquid was removed using a lint-free tissue. Samples were left to air-dry at least 24 h before imaging. TEM images were obtained using Fei Tecnai G2 20 Twin or FEI Tecnai 12 BioTwin transmission electron microscopes (FEI, Hillsboro, USA) at 100 kV beam intensity. TEM images were analyzed by ImageJ software. The scale bar was used to set the scale for the width measurement. A total of 50 measurements of fibers were recorded manually on several images of the same gels to ensure the widths were representative.

### 3.7. Scanning Electron Microscopy (SEM)

A small aliquot (10 μL) of hydrogels was deposited on silicon wafers that were mounted using carbon tape on aluminum SEM stubs and sputtered with a thin gold layer. The samples were observed using a scanning electron microscope (Philips ESEM XL30-LaB6, FEI, Hillsboro, OR, USA) operating at 20–30 kV.

## 4. Conclusions

In the present work we have shown that multicomponent assembly of SAA peptides can form hydrogels with significant differences in the nanotopologies and mechanical properties. The mechanical properties of hydrogels can be influenced by the concentration of the fiber forming peptides where a power law model with a scaling exponent of 5.1 infers the formation of extensive fiber association and bundling. In this report we have further described the possibilities to modify the strength of self-assembling peptide hydrogels by the assembly with different correlated peptides. The tuning of the hydrogel can be realized with simple methods by controlling the ratio of the two different peptides. Diverse results are defined where hydrogels differ in composition, mechanical properties, and morphology. An important aspect playing a key role in the modulation of the hydrogel’s mechanical properties, as well as in the observed nanotopology, could be assigned to branching. Upon growing to sufficient number and length, the fibers of P5 form a 3D network. The presence of the P21 chains modifies the hydrogel’s morphology, introducing branching points that in turn produce dramatic effects on the rheological properties.

In summary, we have shown the spontaneous formation of complex architectures by the assembly of different peptides. The presented study represents an additional step forward to the discovery of new peptide-based nanostructures and how to control their assembly and related properties.

## Figures and Tables

**Figure 1 molecules-27-07901-f001:**
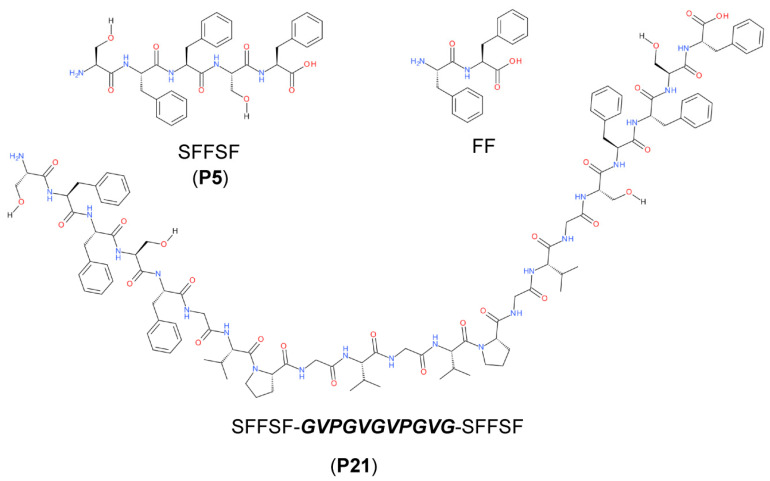
Chemical structures of P5, P21, and FF peptides.

**Figure 2 molecules-27-07901-f002:**
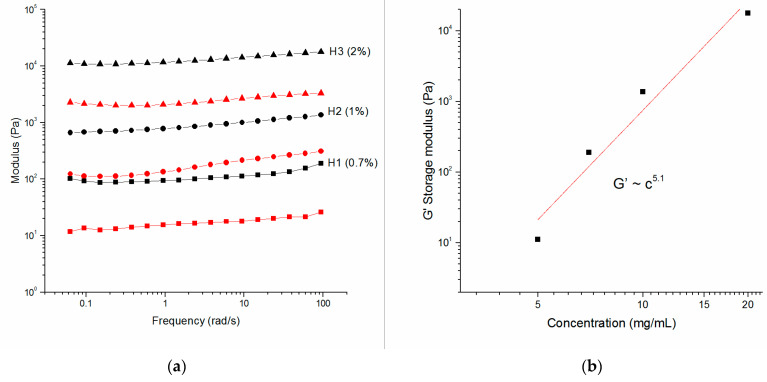
Rheology of hydrogels at different concentration of P5. (**a**) Dynamic frequency sweep at 0.1% of strain of H1 (▲), H2 (●), and H3 (■) hydrogels. Black symbols report storage moduli (G’), while red symbols report loss moduli (G’’); (**b**) storage modulus (G’) at 0.2% strain and 9.4 rad s^−1^ as a function of concentration of P5 peptide.

**Figure 3 molecules-27-07901-f003:**
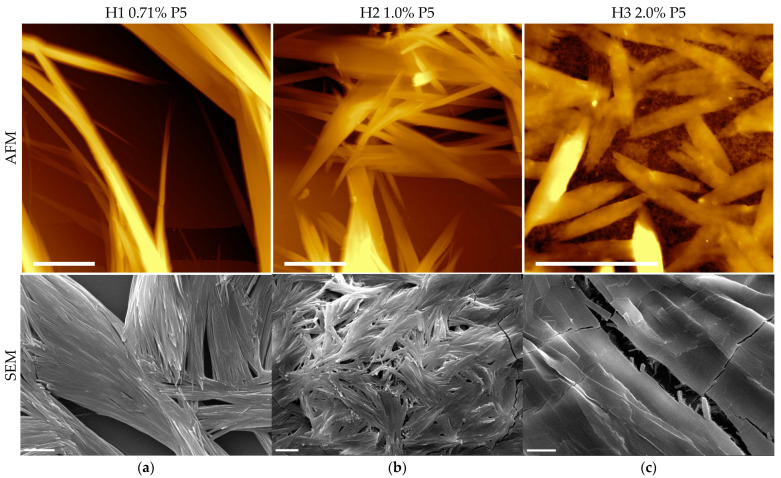
Representative AFM (top) and SEM (bottom) images of hydrogel H1 (**a**), H2 (**b**), H3 (**c**) (scale bar represents 5 μm in AFM images and 10 μm in SEM images).

**Figure 4 molecules-27-07901-f004:**
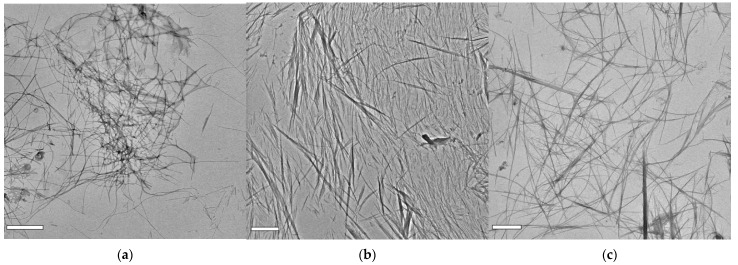
TEM images of diluted samples of hydrogel H1 (**a**), H2 (**b**), and H3 (**c**). The scale bar represents 1 μm (**a**,**c**) and 2 μm (**b**).

**Figure 5 molecules-27-07901-f005:**
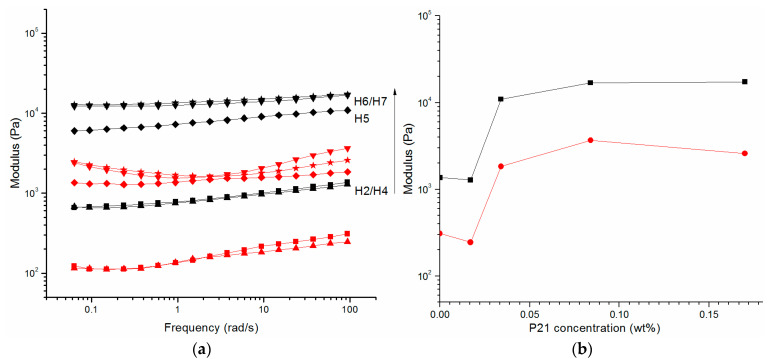
Rheology of hydrogels of P5 1% (*w*/*v*) with different concentration of P21. (**a**) Dynamic frequency sweep at 0.2% of strain of H2 (■), H4 (▲), H5 (♦), H6 (▼), and H7 (★) hydrogels. Black symbols report storage moduli (G’), while red symbols report loss moduli (G’’); (**b**) storage modulus (G’) in black and loss modulus (G’’) in red at 0.2% strain and 9.4 rad s^−1^ as a function of concentration of P21 peptide.

**Figure 6 molecules-27-07901-f006:**
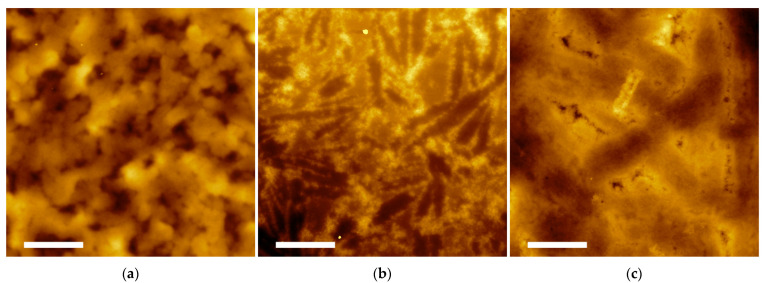
AFM topography images of hydrogel H5 (**a**), H6 (**b**), and H7 (**c**) recorded on samples deposited on silicon wafers after 24 h of gelation. Scale bar represents 5 μm.

**Figure 7 molecules-27-07901-f007:**
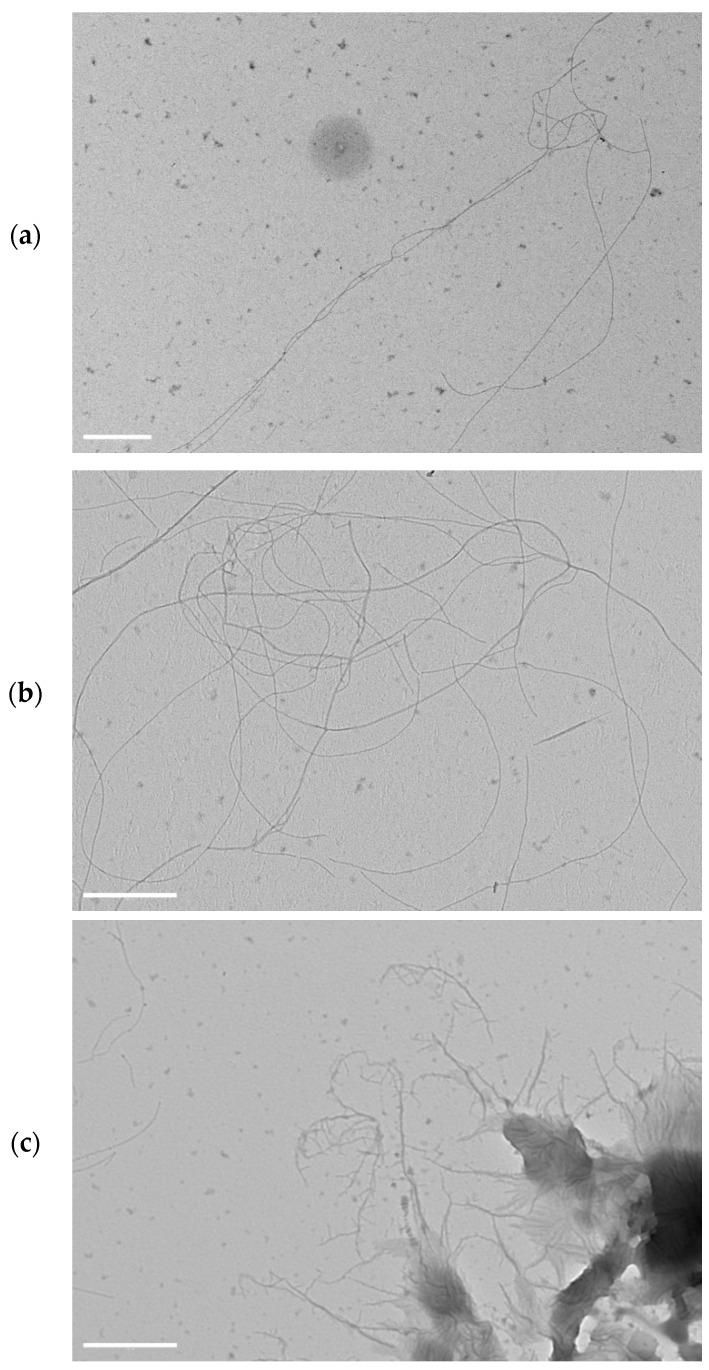
TEM images of diluted samples of hydrogel H4 (**a**), H5 (**b**), and H6 (**c**). Scale bar represents 500 nm.

**Figure 8 molecules-27-07901-f008:**
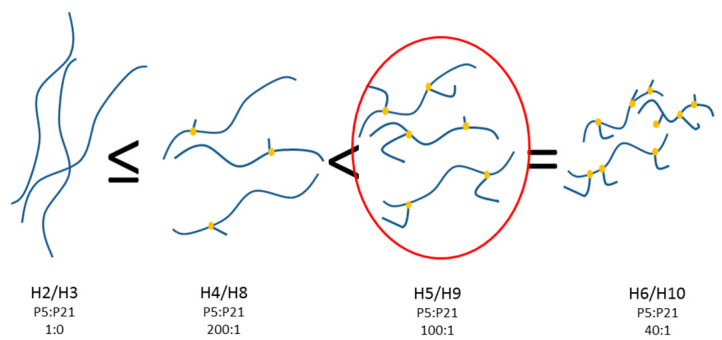
Schematic representation of the hydrogels obtained from assembly of P5 and P21 peptides. In yellow is the P21 peptide representation; in blue are the P5 fibrils. In the red circle, the schematic of P5:P21 (100:1) hydrogels is highlighted.

**Figure 9 molecules-27-07901-f009:**
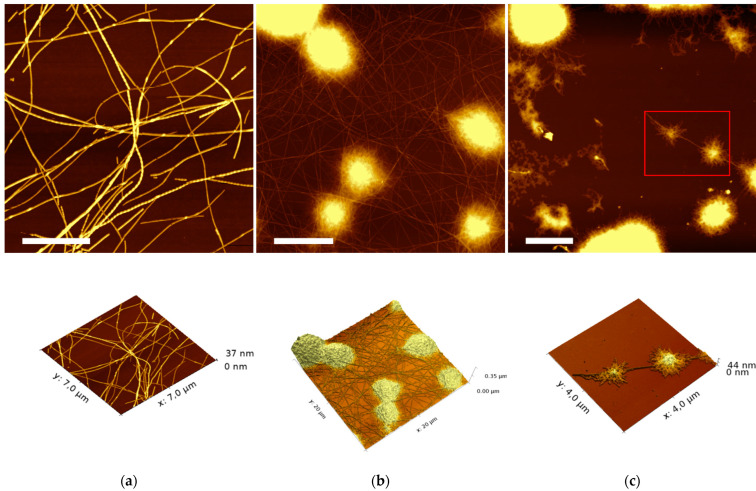
AFM topography images (top) and 3D images (bottom) of hydrogels formed from 1% *w*/*v* P5 and 0% P21 (**a**); 0.25% P21 (**b**), and 0.5% P21 (**c**). Scale bar represents: ((**a**,**c**): 2 μm; (**b**): 5 μm).

**Figure 10 molecules-27-07901-f010:**
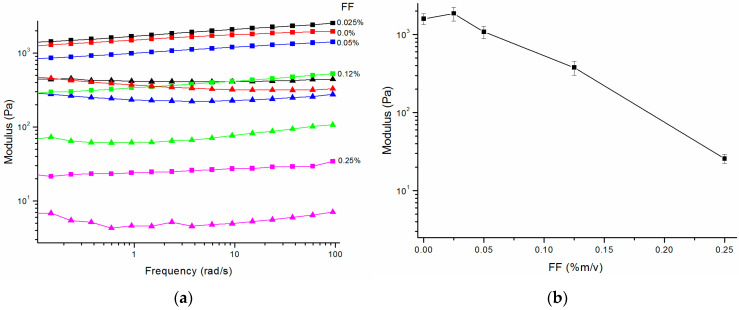
Rheology of hydrogels of 1%P5 (*w*/*v*) with different concentrations of FF nanotubes. (**a**) Dynamic frequency sweep at 0.2% of strain of HF1 (black), HF2 (blue), HF3 (green), and HF4 (magenta) hydrogels. Square (■) symbols report storage moduli (G’), while triangle (▲) symbols report loss moduli (G’’); (**b**) storage modulus (G’) at 0.2% strain and 9.4 rad s^−1^ as a function of concentration of FF nanotubes.

**Table 1 molecules-27-07901-t001:** Composition and mechanical properties of studied hydrogels.

Hydrogel	P5	P21	VIT ^1^	Mechanical Properties	
	(*w*/*v*)	(*w*/*v*)		G’ (Pa)	G” (Pa)
H0	0.5%	-	-	11.2 ± 7.0	5.1 ± 8.4
H1	0.71%	-	✓	111.2 ± 27.5	17.0 ± 3.8
H2	1.0%	-	✓	914.8 ± 224.5	180.6 ± 66.8
H3	2.0%	-	✓	13,226.0 ± 2329.7	2483.2 ± 454.9
H4	1.0%	0.017%	✓	875.6 ± 204.1	162.4 ± 45.5
H5	1.0%	0.034%	✓	8133.2 ± 1638.0	1492.3 ± 178.0
H6	1.0%	0.084%	✓	13,581.5 ± 1491.0	2163.2 ± 644.0
H7	1.0%	0.170%	✓	14,387.0 ± 1495.2	1375.6 ± 318.4
H8	2.0%	0.034%	✓	14,739.5 ± 1962.2	2226.0 ± 404.0
H9	2.0%	0.067%	✓	38,718.0 ± 6066.5	6307.6 ± 1214.6
H10	2.0%	0.17%	✓	39,566.7 ± 5048.0	6114.1 ± 1115.3
H11	2.0%	0.34%	✓	38,072.5 ± 5119.1	5416.1 ± 1040.1

^1^ VIT: vial inversion test.

**Table 2 molecules-27-07901-t002:** Composition and mechanical properties of hydrogels composed of P5 and FF nanotubes.

Hydrogel	P5	FF	VIT ^1^	Mechanical Properties	
	(*w*/*v*)	(*w*/*v*)		G’ (Pa)	G” (Pa)
HF1	1.0%	0.025%	✓	1869 ± 381	426 ± 18
HF2	1.0%	0.050%	✓	1087 ± 198	250 ± 25
HF3	1.0%	0.125%	✓	378 ± 77	75 ± 15
HF4	1.0%	0.25%	-	26.0 ± 3.3	5.6 ± 0.1

^1^ VIT: vial inversion test.

## Data Availability

Data is contained within the article or Appendix A.

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
