# Peer review of "Hydrogels from the Assembly of SAA/Elastin-Inspired Peptides Reveal Non-Canonical Nanotopologies"

_molecules, 2022, doi:10.3390/molecules27227901_

Round 1

Reviewer 1 Report

In the paper “Hydrogels from the co-assembly of SAA/elastin-inspired peptides reveal non-canonical nanotopology”, Alessandra Scelsi and co-workers investigated the co-assembly behavior of different peptides in detail. This work is very interesting and can be published in Molecules after a minor revision.

Comments:

1. Will the second structure of formed fibers be changed along with the increasing of concentration of peptides? Please give CD data.

2. In the experimental section, the authors displayed that different concentration of P5 and P21 can form hydrogels with various mechanical properties. Therefore, how many the peptides take part in the co-assembling process to form fibers? It is better to give the composition of fibers.

3. Did the preparation process of sample of AFM and SEM have effect on the morphology of formed hydrogels? It is suggested to prepare the samples by lyophilization. 

4. Please give the method how to measure the width of fibers.

5. The unit of scale bars in Figure 3 is not clear enough, please revise them.

6. It is better to apply density functional theory (DFT) to simulate the co-assembly of different peptides

Round 2

Reviewer 2 Report

The authors have replied to all my comments and addressed them carefully and properly. I recommend the publication of the reviewed version.